# Targeting Neoantigens in Cancer: Possibilities and Opportunities in Breast Cancer

**DOI:** 10.3390/antib13020046

**Published:** 2024-06-10

**Authors:** Zuhair Chaudhry, Anik Boyadzhyan, Kayvan Sasaninia, Vikrant Rai

**Affiliations:** Department of Translational Research, College of Osteopathic Medicine of the Pacific, Western University of Health Sciences, Pomona, CA 91766, USA; zuhair.chaudhry@westernu.edu (Z.C.); anik.boyadzhyan@westernu.edu (A.B.); kayvan.sasaninia@westernu.edu (K.S.)

**Keywords:** breast cancer, immune response, neoantigens, targeted therapy, personalized therapy

## Abstract

As one of the most prevalent forms of cancer worldwide, breast cancer has garnered significant attention within the clinical research setting. While traditional treatment employs a multidisciplinary approach including a variety of therapies such as chemotherapy, hormone therapy, and even surgery, researchers have since directed their attention to the budding role of neoantigens. Neoantigens are defined as tumor-specific antigens that result from a multitude of genetic alterations, the most prevalent of which is the single nucleotide variant. As a result of their foreign nature, neoantigens elicit immune responses upon presentation by Major Histocompatibility Complexes I and II followed by recognition by T cell receptors. Previously, researchers have been able to utilize these immunogenic properties and manufacture neoantigen-specific T-cells and neoantigen vaccines. Within the context of breast cancer, biomarkers such as tumor protein 53 (TP53), Survivin, Partner and Localizer of BRCA2 (PALB2), and protein tyrosine phosphatase receptor T (PTPRT) display exceeding potential to serve as neoantigens. However, despite their seemingly limitless potential, neoantigens must overcome various obstacles if they are to be fairly distributed to patients. For instance, a prolonged period between the identification of a neoantigen and the dispersal of treatment poses a serious risk within the context of breast cancer. Regardless of these current obstacles, it appears highly promising that future research into neoantigens will make an everlasting impact on the health outcomes within the realm of breast cancer. The purpose of this literature review is to comprehensively discuss the etiology of various forms of breast cancer and current treatment modalities followed by the significance of neoantigens in cancer therapeutics and their application to breast cancer. Further, we have discussed the limitations, future directions, and the role of transcriptomics in neoantigen identification and personalized medicine. The concepts discussed in the original and review articles were included in this review article.

## 1. Introduction

The World Health Organization lists cancer as a leading cause of death worldwide accounting for approximately 10 million deaths in 2020 [1]. Breast cancer is one of the most common forms of cancer accounting for 2.3 million cases and responsible for 685,000 deaths globally [2]. In the United States, the incidence of breast cancer in females is 119 per 100,000 standard population with a mortality rate of 19 per 100,000 standard population [3]. Breast cancer originates from a malignancy of the mammary tissue, primarily in the ducts or the lobules of the breast [4]. Approximately two-thirds of new breast cancer cases are diagnosed at the localized stage (stage 1–2) with a quarter diagnosed at the regional stage (stage 3) and 6% of newly diagnosed breast cancer are metastatic. The prognosis of breast cancer in localized and regional stages is generally favorable as the 5-year relative survival rate ranges from 86% to 99% for the regional and localized stages. Metastatic breast cancer has poor outcomes as the 5-year relative survival rate is at 32% [3,4].

The risk factors for breast cancer include both genetic and non-genetic predispositions. The most common genetic risk factor for breast cancer is a mutation in the BRCA1 and BRCA2 genes [5] (Figure 1). The prevalence of pathogenic BRCA1/BRCA2 mutations is estimated to be 1 in 139 individuals in the general population, with the highest prevalence in Ashkenazi Jewish, Filipino and Southeast Asian, and Non-Ashkenazi Jewish-European populations [4]. Other less common genetic factors for breast cancer include mutations in the *ATM*, *TP53*, *CHEK2*, *PTEN*, and *PALB2* [6] (Figure 1). Genetic profiling has revealed several subclasses of breast cancer based on hormone receptor (ER and PR) and HER2 (ERBB2) status: luminal ER-positive and PR-positive, which is further subdivided into luminal A and B; HER2-positive; and triple-negative breast cancer (TNBC; ER-ve PR-ve HER2-ve) [7]. Nongenetic risk factors for breast cancer include carcinogenic exposure including radiation therapy for other cancers, birth control use, hormone replacement therapy, obesity, age, early menstruation, menopause after 55 years of age, and postponed/absent pregnancy [8] (Figure 1).

Treatments for breast cancer involve a multidisciplinary approach involving a succession or combination of therapies including surgery, radiation, and chemotherapy depending on the type and staging. Mastectomy and salpingo-oophorectomy have been recommended as effective prophylactic measures for individuals with BRCA1/2 mutations [9]. Hormone therapy is the main treatment for hormone receptor-positive breast cancer [10]. Endocrine therapies involve anti-estrogen therapies such as selective estrogen receptor modulators (Tamoxifen), aromatase inhibitors (anastrozole, letrozole, and exemestane), and selective estrogen receptor degraders (fulvestrant) [10]. Chemotherapies such as anthracyclines, taxanes, 5 fluorouracil inhibitors, cyclophosphamide, and carboplatin-targeting pathways involved in cell proliferation are utilized in the treatment of breast cancer [11]. Several therapies were developed targeting the phosphoinositide 3-kinases (PI3K)/protein kinase B (AKT)/mammalian target of the rapamycin (mTOR) pathway. Capivasertibe serves as a protein kinase B (AKT) inhibitor [12]. Alpelisib inhibits phosphoinositide 3-kinases (PI3K) [13], and Everolimus inhibits the mammalian target of rapamycin (mTOR) [14]. Palbociclib, ribociclib, and Abemaciclib are CDK4/6 inhibitors inhibiting the cell cycle [15] (Figure 1).

Surgical intervention is invasive and this makes it incurable by simple surgery in advanced stages of breast cancer [16,17]. Additionally, off-target effects due to the systemic nature of endocrine, chemotherapy, and radiation therapy provide challenges in the management and treatment of breast cancer [18,19]. Furthermore, selective pressures in the tumor microenvironment, in addition to dysregulated DNA editing mechanisms in tumors following chemotherapy and hormone therapy contribute to loss or gain of function mutations enabling resistance to treatment [20]. Combined with the high mortality rate of metastatic breast cancer, there is a high need to investigate novel treatment modalities in the treatment of resistant metastatic breast cancer.

## 2. Current Treatment and Resistance to Therapy

Recent attention is on developing therapies that utilize the host immune system for the treatment of breast cancer. Unlike non-small cell lung cancer, melanoma and glioblastoma where immune checkpoint blockade therapy was first FDA approved, the efficacy of immune checkpoint inhibitors in breast cancer is limited [21]. In normal immunity, T-cells are activated by antigens in the context of a major histocompatibility complex in coordination with co-receptor binding to induce tumor cell clearance [22]. To prevent T-cell activation in host tissues, immune checkpoints are involved to induce T-cell tolerance [23]. The program death cell receptor, PD-1, expressed on T-cells interacts with PD-1 ligand PD-1L expressed on antigen-presenting cells to induce T-cell inactivation [24]. Neoantigens, the cancer-specific antigens developed due to mutations, are bound in the binding groove of MHC of APCs and are presented to T cells [25]. The PD-1/PD-1L axis is one of the primary mechanisms of breast cancer evasion of host immunity [26]. The overexpression of PD-1 ligand on breast cancer tissue is associated with increased tumor size, cell proliferation, HER2 positive status, negative estrogen receptor status, and breast cancer mortality [26]. Immunotherapy was developed to direct and augment cell-mediated responses to cancerous cells. Immunotherapies have been approved by the FDA for the treatment of breast cancer targeting the PD-1/PD1-L axis [27]. Pembrolizumab and Dostarlimab are PD-1 inhibitors and Atezolizumab is a PD1-L receptor inhibitor [28].

Similar to chemotherapy and hormonal therapy, treatment resistance also occurs for immune checkpoint inhibitors [29]. Immune checkpoint resistance mechanisms include tumor-mediated mutations, the upregulation of inhibitory immune checkpoints, and an increase in immunosuppressive factors in the tumor microenvironment [30]. Interferon (IFN)-γ overexpression in addition to increased Janus kinase (JAK)-signal transducer and activator of transcription (STAT) signaling induces increased PD-1 ligand expression and leads to CD8 T cell exhaustion [31]. A decrease in MHC expression and antigen presentation can lead to the decreased efficacy of immune checkpoint inhibitors due to the lack of T-cell activation [32]. PD-1/PD-L1 inhibition can also lead to an increase in compensatory immunosuppressive proteins such as upregulation in lymphocyte activator gene 3 (CD223), T cell immunoglobulin 3 (CD366), V-domain Ig inhibitor activated by T cells (VISTA), and T cell immunoglobulin and ITIM domain protein (TIGIT) [30]. Increased immunosuppressive modulator expression, such as transforming growth factor (TGF)-β, promotes Treg cell expansion and proliferation [33]. Changes in the tumor microenvironment with increased vascular endothelial growth factor (VEGF) and angiogenesis can dysregulate T-cell activation and antigen presentation [34]. Extracellular matrix remodeling induced by changes in the tumor microenvironment can lead to tumor cell expression of extracellular proteinases and inhibit T-cell transport to tumor cells [35].

Neoantigens are a class of proteins that are differentially expressed in cancer cells, absent in healthy cells, and can induce a cell-mediated response [36,37]. While neoantigens play a critical role in the efficacy of PD-1/PD-1L checkpoint inhibitors, resistance can occur due to an active antitumor response resulting in immunoselection and outgrowth of neoantigen-loss variants and tumor immune escape y [38]. Thus, therapies that can obviate the factors leading to treatment resistance are needed. Current research investigates expanding the repertoire of neoantigens for identification in addition to using neoantigens as direct targets for therapy. Neoantigen-based immunotherapies under investigation include adoptive cell therapy, vaccines, and bispecific antibodies [36]. Adoptive cell therapy utilizes T-cell receptor (TCR) adoptive therapy or chimeric antigen receptor (CAR) T cells manufactured to selectively recognize specific neoantigens [39]. Vaccines use neoantigens peptides or nucleic acid sequences to create immunologic memory to prevent the onset of cancer [40]. Bispecific antibodies have a neoantigen binding domain and a T-cell binding domain that assists T-cells to the tumor site [41]. Notable neoantigens as potential therapeutic targets in breast cancer include TP53, survivin, PALB2, and PTPRT (Figure 1B–E). This review aims to comprehensively discuss the current landscape and limitations of breast cancer neoantigen research and neoantigens as therapeutic targets.

## 3. Neoantigens

Cancer immunotherapy has revolutionized cancer treatment by harnessing the power of the immune system to target tumor cells (Figure 1A). However, still there are various concerns in immunotherapy resulting in resistance and recurrence. Thus, there is a need for a novel target which can effectively be targeted to treat the cancer. Neoantigens arising from somatic mutations in tumor cells have emerged as key players in this context offering promising avenues for personalized cancer therapy. Recent advancements in cancer immunotherapy have made it possible to navigate immune responses to patient-specific neoantigens. The recognition of these neoantigens is believed to be a major factor in the development of immunotherapies [42]. Thus, understanding the role of neoantigens in therapy resistance and cancer recurrence not only in the context of breast cancer but also in other cancers is critical for optimizing treatment strategies and improving patient outcomes.

Neoantigen refers to antigens that result from somatic mutations in the tumor cells generating mutant peptides distinguished from wild-type peptides presented by major histocompatibility (MHC) molecules in tumor cells. Antigens/neoantigens are processed intracellularly (antigen processing), and antigenic peptides are bound to MHC molecules. The production of these tumor-specific antigens (TSAs) is caused by genetic alterations including single nucleotide variants (SNVs), structural variants (SVs), and posttranslational modification in the genome, transcriptome, and proteome [43]. The predominant form of mutation leading to the generation of mutant peptides within tumor cells at the genomic level is SNVs. Due to their inherent characteristics, SNVs exhibit significant variation both across stages within a particular cancer and among different types of cancers. Given that mitochondria are present in the tumor cells and play a crucial role in tumor metabolism and metastasis, SNV inducing in TSAs has also been observed in mitochondrial DNA (mtDNA), resulting in the production of numerous mutant peptides [44].

Insertion and deletion (INDEL) mutations are caused by the insertion or deletion of nucleotide base pairs, often causing frameshift mutations. Due to the systemic change of the genome caused by INDELs, they can produce more neoantigens than SNV with increased MHC-I binding affinity [36]. Neoantigens resulting from frameshift mutations are not only more prevalent in cancers with high microsatellite instability (MSI) than those derived from SNVs but also exhibit a robust correlation with immune responses. Thus, neoantigens represent promising avenues for immunotherapy directed towards MSI cancers such as colorectal, endometrial, and even breast cancer, cancer that has been studied to relate to MSI [45]. Roudko et al. [46] showed that based on clinical phase I/IIa trials, neoantigen vaccines encoding frameshift mutations exhibit favorable tolerance and consistently induce immune responses. These results imply that neoantigen-based vaccines are viable choices for addressing MSI cancers in both treatment and prevention.

In nonpathological cells, RNA splicing is the conversion from immature mRNA to mature RNA; however, it can be influenced by mutations in cis/trans-regulatory factors and core spliceosomes. These abnormal splicing patterns are another source of neoantigens. Mutations occurring within cis-acting elements contribute to the generation of potential neoantigens by altering splicing processes like intron retention and exon skipping. In almost all cancer types, intron retention is more prevalent compared to normal tissues. Although transcripts with retained introns undergo nonsense-mediated RNA decay (NMD), neoantigens can still emerge from intron-retained transcripts before their degradation. The generation of neoantigens from skipped exons is referred to as neojunctions and is more frequently compared to those derived from SNV mutations [47]. As tumor-specific exitron-spliced transcripts possess exitrons—exon-embedded cryptic introns with both splicing and protein-coding potential but lacking stop codons—they are more prone to evade NMD. Consequently, their overall expression surpasses that of retained introns [48]. Mutations occurring in trans-acting elements result in an altered splicing variant due to a somatic mutation in a splicing factor, thus inducing the production of neoantigens as well [36]. Neoantigens derived from these variants have driven the development of therapeutics in tumors [49] where mutated splicing factor 3b protein complex subunit 1 (SF3B1), a splicing factor found in the spliceosome, is mutated in uveal melanoma, it produces neoantigens specific to the tumor. These neoantigens trigger CD8+ T cells, leading to the destruction of tumor cells. The current research suggests similar mechanisms may be present with additional cancers including breast cancer.

## 4. Resistance—Neoantigen Loss and Defects in Presentation

Identifying TSAs provides a key to tumor-specific therapies; however, it has been recently discovered that cancer cells have evolved resistance to this anti-cancer immune response. Further, a loss of tumor cell neoantigen expression can occur over time due to the overall reduced expression of tumor-causing genes or loss of the neoantigen-specific T cell reactivity in melanoma [50]. The altered immune response in tumors with loss of neoantigens targeted by T cells results in the growth of tumor cells [38] and this loss of neoantigens can be induced by the immune system. It was found that in tumors with less immune cells, there was a decrease in the elimination of neoantigens during the cancer’s development. Whereas tumors with more immune cells have shown an ongoing elimination of neoantigens either through changes in immune cell genes or the reduced expression of neoantigens. Epigenetics also plays a role in the silencing of neoantigenic mutations via promoter hypermethylation. These findings suggest that the immune system plays a significant role in the development of immune evasion by cancers [51].

In addition to neoantigen loss, tumor cells may have defects with antigen processing and presentation, which allow neoantigens to evade immune responses. To determine the frequency of this phenomenon in response to anti-programmed cell death (PD) therapy, a study utilized a computational tool to detect the loss of diversity in human leukocyte antigen (HLA). In non-small cell lung cancer, McGranahan et al. [52] found that 40% of the cancers had HLA loss through heterozygosity, leading to the loss of binding with neoantigens and evasion of the immune system. It is unclear whether HLA loss of heterozygosity prevents presentation during anti-PD therapy. This immune evasion due to defects in antigen presentation may also apply to therapies used in breast cancer.

## 5. Tumor-Specific Antigens (Neoantigens) in Breast Cancer

Breast cancer is a malignancy originating in the mammary glands. Understanding the role of neoantigens, which are unique markers on cancer cells, holds promise for developing targeted therapies and immunotherapies in breast cancer. Current research on biomarkers for breast cancer including tumor protein 53 (TP53), Survivin, Partner and Localizer of BRCA2 (PALB2), and protein tyrosine phosphatase receptor T (PTPRT) have shown promising avenues for identifying and treating breast cancer [53].

### 5.1. TP53

Tumor-associated antigens entail a plethora of molecules and are thus divided into various categories based on their patterns of expression. *TP53* falls within the category of “Over Expression Antigens” as it is a gene whose resulting proteins are found in normal tissue but become overexpressed in cancer [53] due to mutation. For context, TP53 encodes for p53, responsible for transcribing genes dealing with many processes including metabolism, DNA repair, and cellular senescence after exposure to stressors like chemotherapy or radiation. With regard to breast cancer, mutations of the *TP53* gene remain the single most common genetic alteration associated with tumors [54]. The stark difference in the degree of the expression of the proteins encoded by p53 within healthy tissue compared to that of cancerous tissue makes it a strong contender for cancer immunotherapy. Umano et al. [55] investigated the potential of p53-encoded HLA-A24 binding peptides in eliciting anti-tumor cytotoxic T lymphocyte (CTL). Within the study, the HLA-A24 peptide binding motifs were presented by molecules H-2Kd and H-2Kb to cytotoxic T lymphocytes as a result of their similarities. Researchers were able to pinpoint the peptide p53–161 and ascertain its ability to initiate an immune response from CD8+ T cells, which would then effectively destroy tumor cells that expressed both HLA-A24 and p53. The study concluded that HLA-A24-positive patients can benefit from treatments targeting p53 mutations (Figure 1).

### 5.2. Survivin

Survivin is an essential protein involved in normal cell growth and division, showing a marked increase in expression during the G2/M phase of the cell cycle. In addition, survivin has been shown to directly inhibit apoptosis upon binding to caspases and preventing their activation. Importantly, this protein is not found within normal tissue and is expressed heavily within most neoplasms; for instance, survivin is present in the tissues of roughly 65% of the patients with breast cancer. More specifically, those patients diagnosed with ductal carcinoma in situ (DCIS) and TNBC [56]. Survivin may be co-expressed with various proteins associated with breast cancer such as HER 2 and Urokinase Plasminogen Activator [56]. Survivin has the potential to offer a novel treatment for breast cancer because of its therapeutic efficacy in patients with chronic lymphatic leukemia and melanoma. Specifically, epitopes acquired from survivin were shown to initiate a cytotoxic T lymphocyte response in leukemia (three out of four) and melanoma (three out of six) patients. In contrast, no such response was recorded within the healthy controls, highlighting the potential of survivin as a future neoantigen in cancer treatment [57]. Promising outcomes such as these have recently inspired the analysis of survivin expression levels within women diagnosed with DCIS. The next-generation sequencing and histopathological studies of tumor and control tissues suggesting the overexpression of wild-type survivin in the tumor tissues provide clear evidence of developing new CTL immunotherapy treatments for DCIS [58]. Thus, targeting survivin will promote apoptosis increasing death and clearance of tumor cells (Figure 1).

### 5.3. PALB2

Partner and Localizer of BRCA2 (PALB2) is situated on chromosome 16p12.2 [59] and is accountable for the nuclear localization of BRCA2 and the repair of DNA damage [60]. It plays an important role in DNA damage repair through two linked pathways: Fanconi anemia (FA) and homologous recombination (HR) [61]. FA is a rare genetic instability syndrome caused by biallelic pathogenic variants in FA genes, leading to early onset bone marrow failure and increased cancer susceptibility [62]. PALB2 falls under group 3 proteins of the FA pathway, acting as a downstream effector to facilitate rapid or DNA interstrand cross-links [63]. As a crucial tumor suppressor gene involved in the homologous recombination (HR) pathway, PALB2 acts as a mediator between BRCA1 and the BRCA/RAD51 complex. Similar to individuals with mutations in the BRCA1/2, the inactivation of both PALB2 alleles results in the initiation of nonhomologous end joining (NHEJ), leading to genomic instability and the production of foreign proteins [64,65]. Zhang et al. [66] used next-generation sequencing to identify cancer neoantigens and found that PALB2 induced significant peptide-specific T-cell responses. Furthermore, to evaluate the role of neoantigen-specific CD8+ T cells in tumor immunity, they implanted WHIM30 tumor sections into immune-compromised mice and adoptively transferred autologous PBMCs stimulated with PALB2, ROBO3, or CMV peptides. Results showed that adoptive transfer of PBMCs stimulated with PALB2 and ROBO3 leads to reduced tumor growth, while CMV-stimulated PBMCs had no effect, displaying the viability of using neoantigens like PALB2 to induce antitumor immunity [66].

### 5.4. PTPRT

Protein Tyrosine Phosphatase Receptor Type T (PTPRT), belonging to the type IIB RPTP subfamily, comprises an extracellular domain (including a meprin/A5/PTP μ domain, an Ig domain, and four fibronectin type III repeats), a transmembrane domain, a juxtamembrane region, and two phosphatase domains (D1 and D2) [67]. It plays a critical role in suppressing tumor growth and enhancing cell adhesion across various cancers. *PTPRT* is a tumor suppressor gene regulating cell cycle and cell adhesion. Notably, five missense mutations in the frequently altered PTPRT have been shown to decrease phosphatase activity. The expression of wild-type PTPRT, but not mutant forms, inhibited cell growth in human cancer cells [68]. Additionally, Zhang et al. [69] demonstrated that deletion of the fibronectin type III repeats (FNIII) of PTPRT resulted in impaired cell/cell aggregation, suggesting that PTPRT inactivation may promote cancer progression by disrupting cell/cell adhesion [69]. To study the role of PTPRT in breast cancer, a recent study found that the mRNA level of PTPRT could be used as biomarkers for different stages, age groups, and grades of breast cancer. The study showed that larger tumors were associated with a lower PTPRT expression level and a low proliferation rate with a high PTPRT level [70]. This pathology can be due to the production or lack of neoantigens. Chen et al. [71] used the human leukocyte antigen (HLA) allele from patients and selected the recurrent mutations of PTPRT for neoantigen prediction and found that some recurrent mutations in p.G826R and p.R1117C of PTPRT were predicted to generate high-affinity neoantigens that could be used as potential targets for immunotherapies.

## 6. Neoantigens as a Novel Treatment in Breast Cancer

Studies highlight how the relative novelty of breast cancer neoantigens presents an obstacle in designing new treatments. For instance, Toss et al. [72] highlighted that previous studies focused on PALB2-based immunotherapy but most of these studies failed to support its efficacy. Instead, they highlighted a study based on five independent cohorts that followed 672 patients diagnosed with advanced melanoma. In all cases, the PALB2 mutation was associated with a higher mutation rate as well as a neoantigen burden level. Patients with the PALB2 mutation showed an improved objective response rate and overall survival compared to their wild-type counterparts. Ultimately, Toss et al. conclude that PALB2 can be utilized as a positive predictor in immunotherapy treatments based on CTLA4 inhibitors targeting melanoma [72]. More promising findings have been found when assessing neoantigens derived from a mutation in PTPRT. For example, the neoantigen arising from mutation G826R can successfully elicit a response from HLA-B44:02 and thus be transformed into a cancer vaccine [71].

Neoantigens have been utilized to produce different forms of vaccines via a highly standardized process. A few of these vaccines are detailed below alongside their efficacy based on previous studies not limited to breast cancer.

Tumors are often accompanied by neoantigens, a product of their genetic instability and high mutation rate. Neoantigens are unique from the antigens typically expressed in normal tissue because they elicit an immune response and are considered highly immunogenic. Researchers take advantage of this immunogenic property, as well as the varying affinity that neoantigens have towards major histocompatibility complexes (MHC) via whole-exome sequencing [40]. Whole-exome sequencing is conducted on a tumor biopsy and the adjacent noncancerous tissue, pinpointing neoantigens recognized by MHC classes I or II in the process. These selected neoantigens are then tested in vitro and in vivo regarding their ability to initiate an immune response via CD4+ and CD8+ T cells, the latter being weighed more heavily [73]. During this stage, researchers utilize software programs that help analyze the immunogenicity of antigens, using the resulting data to select an “optimal tumor neoantigen” [40]. However, even after identifying a possibly immunogenic neoantigen, one must ensure that it is efficiently presented by an MHC molecule and recognized by a T cell receptor (TCR). Satisfying the former requirement does not necessarily ensure that the second will be met. Many potential neoantigens that are presented by an MHC molecule do not induce an immune response such that one cannot neglect TCR recognition, which can be assessed by in silico techniques [36]. Upon identification, researchers can utilize the immunogenic properties of selected neoantigens to generate new forms of tumor immunotherapy. These therapies can fall under one of two classifications: neoantigen-specific T-cells and neoantigen vaccines [74]. Currently, vaccines target neoantigens including nucleic acid, dendritic cells, synthetic long peptide-based vaccines, and other targets (Figure 1). Despite their differences, these vaccines have been shown to provide better results when co-administered with other treatments such as immune checkpoint therapy, radiation therapy, and chemotherapy [40].

Nucleic Acid vaccines: The mechanism of a DNA vaccine is well studied and has been shown to treat numerous pathologies successfully since the 1990s. Upon closer investigation, researchers discovered the selective advantages of these vaccines, which can positively impact the course of cancer treatment and prevention. DNA vaccines can be administered by intramuscular, intradermal, subcutaneous, or mucosal routes after which they are translocated into the nucleus of a cell [73]. Within the nucleus, the vaccine will utilize bacterial plasmids to manufacture antigens. These antigens are unique in that they are nonspecific to HLA types, meaning that they may be presented by both MHC classes I and II. This directly enables the initiation of a broader cellular immune response; in addition, the recruitment of B lymphocytes allows for antigens to simultaneously trigger a humoral immune response. Within the setting of cancer treatment, these properties give DNA vaccines the potential to prevent metastasis as they stimulate a systemic immune response. The therapeutic capabilities of DNA vaccines have been well established within a few published studies, such as the phase II clinical study, NCX01304524. The study enrolled patients suffering from cervical intraepithelial neoplasia, and intramuscularly administered 6mg of the vaccine VGX-3100 once per month for three months. Upon analysis, the vaccine was proven to be efficacious in half of the enrolled patients, as evidenced by an increase in the number of CD8+ T cells and B lymphocytes compared to the control group that received a placebo drug. Due to these results, VGX-3100 is now recognized as the first therapeutic vaccine to initiate a complete adaptive immune response in patients diagnosed with preinvasive cervical disease following infection by HPV-16 and 18 [73].

Dendritic cell vaccines: Dendritic cell vaccines have garnered a lot of attention due to their well-known success in the treatment of various solid tumors and melanomas. In recent clinical trials, researchers were interested in the potential benefits of neoantigen-based therapy for the treatment of lung cancer, specifically utilizing personalized neoantigen peptide-pulsed autologous DC vaccine, also known as Neo-DCVac [75]. In total, 12 patients suffering from metastatic lung cancer were included in the clinical trial, and each was assigned 12–30 peptide-based neoantigens. These neoantigens were selected using the whole-exome sequencing technique mentioned earlier to locate various somatic nonsynonymous mutations. Researchers not only focused on neoantigens recognized by MHCs but also added a layer of complexity by analyzing the human leukocyte antigen (HLA) haplotypes [75]. This was an essential step as the HLA class I and II alleles helped determine which neoantigens would be presented to T lymphocytes. Finally, RNA sequencing helped isolate any fusion-based neoantigens, a type of mutation found within tumors due to either mesenchymal deletions or the alterations of chromosome structure such as translocation or inversion [36].

Eventually, dendritic cells were manufactured to create the Neo-DCVac, such that a median of five doses were administered per patient. Ultimately, the effectiveness rate was 25% and the disease control rate was 75%, the median overall survival was roughly 8 months. It is important to note that Neo-DCVac, similar to other vaccines targeting neoantigens, works best when administered in concert with other treatment modalities such as adoptive cell therapy (ACT) and immune checkpoint inhibitors (ICI). For instance, the clinical trial enrolled four patients who were also being treated via ICI but either saw no response or suffered a relapse. Once NEo-DCVac was coadministered, these four patients successfully controlled their disease with an 80% reduction in tumor size [75].

Peptide vaccines: Peptide vaccines are heavily researched as a potential avenue for personalized neoantigen vaccines since they are safe and highly specific. Due to their chemical properties, peptides may be deliberately manufactured to be short, long, or even fused before undergoing various chromatographic purification processes. Afterward, polypeptides are combined with an adjuvant and injected subcutaneously into patients, which has proven a safe and effective treatment modality [36]. For instance, the results of a phase I immunotherapy trial focused on patients suffering from disseminated synovial sarcoma demonstrated positive outcomes in five subjects [76]. The administration of an SYT-SSX peptide vaccine successfully stopped disease progression in one of the subjects, while initiating a cytotoxic T lymphocyte (CTL) response in the four other subjects. Despite different responses, no serious adverse effects were reported in the five patients treated [76]. Studies such as these support the application of synthetic peptide-based neoantigen vaccines for the treatment of multiple types of cancers including bladder, colorectal, pancreatic, and breast cancer. These results suggest the beneficial effects of neoantigen-based treatments; however, there is a need for novel antigens for personalized therapy.

As discussed above, targeting neoantigens seems to be a beneficial therapeutic strategy for cancer. The neoantigen-based vaccines have been applied to various other forms of cancer with success and this suggests that they may be used as a model to generate future breast cancer treatments. The findings from both finished and ongoing clinical trials investigating neoantigen-based treatment for breast cancer, as summarized in Table 1, underscore the significance of directing therapeutic efforts toward neoantigens in breast cancer.

The clinical trials listed in Table 1 indicate that tumor heterogeneity plays a critical role in treatment type and one treatment type may not be effective in another type. Breast cancer is genetically and clinically heterogeneous and luminal A, luminal B, HER2-positive, and triple-negative (estrogen -ve, progesterone -ve, and HER2 -ve) are common subtypes based on the viewpoint of the treatment. The small portion of breast cancer cells resistant to immunotherapy, radiation therapy, and chemotherapy may cause treatment failure, recurrence, and metastasis [77,78] and may be the underlying reason for the effectiveness of one treatment while not of another in a particular subtype of breast cancer. Thus, it is essential to investigate neoantigens specific to tumors in a patient and treat them accordingly to promote personalized therapy with improved clinical outcomes. The different responses to treatment mainly pertain to the presence of cancer stem cells, a specific subpopulation of tumorigenic cells which can differentiate in large heterogenous tumor cell subpopulation with a different phenotype via differentiation hierarchy making tumor cells resistant to treatment, propagate, and metastasize [79]. The presence of multiple cell types in the cancer makes it difficult to target multiple genetic subpopulations effectively and thus, the identification of specific neoantigens seems to be beneficial to improve clinical outcomes.

## 7. Recent Advances in Neoantigen Detection

Neoantigens may be identified using immunogenic or immunopeptidomic approaches [80]; however, for personalized medicine, more recent techniques should be used to identify neoantigens. Neoantigens prediction requires several steps: (i) the extraction and whole exome sequencing (WES) of the tumor tissue sample, (ii) somatic variant detection through pairwise alignment of normal and tumor DNA sequences, (iii) the analysis of tumor gene expression (iv) TCR profiling and HLA haplotyping, and (v) variant protein and epitope prediction [81]. However, there are current limitations in workflow duration and cost, in addition to challenges in narrowing therapeutic neoantigen targets. A variety of methods have been explored to increase the efficiency and specificity of neoantigen detection in breast cancer. Hashimoto et al. evaluated the utility of using RNA-sequencing (RNA-seq) data from six breast cancer patients in place of WES to detect neoantigen candidates [82]. Utilizing RNA-seq data alone generated a higher false positive and false negative rate than WES data. However, a combination of tumor RNA-seq data and normal WES data detected neoantigen candidates that have higher expression and rich variant transcripts than WES data alone. Nguyen et al. [83] have also explored using RNA-seq data from 25 colorectal patients to predict neoantigen candidates. In contrast to the Hashimoto et al. study [82], utilizing RNA-seq data alone significantly increased the number of highly immunogenic validated neoantigens detected compared to utilizing DNA seq data [83].

Tretter et al. [84] explored a multi-omics approach in creating a pipeline for neoantigen detection using proteogenomic data from 32 patients across 25 cancer types. The study combined DNA and RNA sequencing with liquid chromatography-tandem mass spectrometry of immunoprecipitated HLA class I peptides in their pipeline. A broad diversity of non-canonical HLA-binding peptides was detected with this approach; however, the authors noted that the majority of neoantigens detected originated from variants identified from the RNA-seq data, also pointing to the transcriptome as a source for novel neoantigen detection. Animesh et al. [85] also explored a multi-omics approach combining WES and RNA-seq data to assess the neoantigen burden of breast carcinoma on seven breast cancer patients. A report from their preprint demonstrated their pipeline was able to predict 434 unique neoantigens from 237 different genes with 87% of the neoantigens expressed at the RNA level. Additionally, 99.98% of the novel neoantigens were patient-specific and infrequently shared between patients. Furthermore, most of the neoantigens were MHC class I specific and are rarely shared between MHC class I and MHC class II [85].

To identify neoantigens with broad immunogenicity across patients, Mistretta et al. developed a framework to detect neoantigens from chimeric RNA sequences resulting from a fusion of two separate genes in cancerous tissue [86]. The authors mined paired-end RNA sequence reads from 75 breast tumors of TNBC, HER2+, and HR+ subtypes, analyzed binding affinity to MHC I through MHCnugget binding predictor, and verified immunogenicity through in vitro enzyme-linked immunospot (ELISpot). The study revealed 20 novel fusion transcripts with further exploration of the NSFP1-LRRC37A2 fusion transcript. Further exploration demonstrated that 15 different neoantigen peptides were discovered from NSFP1 and LRRC37A2 truncations and were predicted to bind onto 35 unique MHC I alleles. A number of computational strategies have been developed to predict fusion neoantigens from RNA-seq data [87,88,89,90]. Overall, these findings indicate that RNA-seq data from healthy and breast cancer tissue is a central component of novel neoantigen detection and efforts to streamline the process are under investigation. However, it should be noted that in silico analysis has a limitation because the identified neoantigen is based on machine learning and their performance is database-dependent that was used to identify [91].

## 8. Future Direction

Therapies targeting neoantigens represent a promising future in the treatment of breast cancer, but many restrictions still exist and are discussed throughout this section. By prioritizing research initiatives that seek to overcome these restrictions, our ability to generate and distribute these highly personalized treatments will be vastly improved. Discussions from multiple clinical trials often highlight the difficulty of determining neoantigens using current resources. Sequencing techniques have been proven to identify neoantigens with a relatively low rate of false negatives. However, one major flaw persists: only a small fraction of these neoantigens are successfully recognized by T cell receptors (TCRs), meaning that the vast majority fail to elicit an immune response. Rejecting unrecognizable neoantigens is essential for the future of personalized treatments, especially for tumor lines with high mutation rates [7]. Therefore, the creation of tools capable of filtering the cancer exome of nonimmunogenic neoantigens represents one potential avenue for increasing the therapeutic efficacy of these treatments. In contrast, researchers can focus on improving existing tools such as NGS technology and mass spectrometry to identify new neoantigens. It is posited that these neoantigens will most likely arise from DNA segments of the human genome with unknown functions, also known as “dark matter” [44]. Another important consideration deals with the timing of these therapeutic modalities given the severity of various diagnoses, including those pertaining to breast cancer. Currently, the process of identifying a neoantigen and generating a personalized vaccine lasts about 4.5 months, such that time becomes a major risk factor for patients suffering from metastases [40]. Modifying this process to create an initial vaccine sooner will significantly benefit patient outcomes as they would start treatment sooner. In the months following initial vaccination, boosters should be administered according to the patient’s diverse needs. During this time, physicians must assess the outcomes of different combination therapies, specifically prioritizing those that minimize immunosuppression within the tumor microenvironment [44].

Many of the limitations that accompany neoantigen-based therapy arise because it is a new treatment option. As researchers continue to perform more clinical trials, the process of locating neoantigens, and generating treatments will become more time efficient and cost-effective. Monetary restrictions are currently a large burden to accessibility as most patients cannot afford these novel treatments.

Unfortunately, immunotherapies targeting breast cancer are not immune from these general limitations as evidenced by the studies conducted by Hashimoto et al. Throughout the study, researchers examined six patients diagnosed with hormone receptor-positive and human epidermal growth factor receptor 2-negative invasive carcinoma. The results showed that RNA sequencing alone was an unreliable method for selecting neoantigens due to high false-positive and false-negative rates. However, it is noteworthy that combining RNA sequencing with whole exome sequencing proved to be more promising as a means of detecting neoantigens before producing a vaccine [76].

These obstacles do not minimize the impact neoantigens will make on the future of cancer treatment, especially regarding the everlasting challenge of drug resistance. More specifically, researchers have developed the ability to predict changes that neoantigens will undergo throughout a patient’s treatment, which coincide with the evolution of a drug-resistant tumor [41]. Exploiting this speculative nature can enhance the efficacy of neoantigen-based therapy used independently and in concert with other forms of immunotherapy such as chimeric antigen receptor (CART) T cells. CAR T cells are founded upon two basic principles: the specificity of antibodies towards antigens as well as the effector function of T cells. As of now, the capacity for CAR T cells to treat solid tumors is limited due to the heterogeneity associated with breast cancer and antigen loss, both of which enable tumor escape [92]. Difficulty in selecting precise targets in CAR T cell therapy generated prominent levels of toxicity. It can be posited that successfully uniting the aforementioned traits of neoantigens with the deployment of cytokines via CAR T cells can overcome the suppressive nature of solid neoplasms, thus reducing the mortality rate associated with triple-negative breast cancers. These unique qualities of neoantigen-based treatments further solidify their budding role in the future of breast cancer detection, treatment, and prevention.

## 9. Conclusions

The landscape of breast cancer treatment is evolving with a shift towards understanding and leveraging neoantigens. This emerging field offers a promising avenue for improving patient outcomes by harnessing the immune system’s potential to target tumor-specific antigens. Despite challenges such as timely delivery and widespread accessibility, the exploration of neoantigens as therapeutic targets holds immense potential. Biomarkers like TP53, Survivin, PALB2, and PTPRT exhibit promising immunogenic properties, suggesting a path toward personalized and targeted therapies. The breast tumor heterogeneity intrinsic to the cell-of-origin has a large impact on antigen and neoantigens load and type which cancer cells exhibit on their surfaces and this poses difficulties in targeted personalized therapy. Neoantigens, the ideal targets for immunotherapy, should generate a specific reaction towards a tumor antigen for improved clinical outcome. While further research is warranted to overcome existing hurdles, the trajectory of neoantigen-based therapies signifies a transformative approach in the fight against breast cancer, offering hope for enhanced treatment modalities and improved clinical outcomes in the future.

## Figures and Tables

**Figure 1 antibodies-13-00046-f001:**
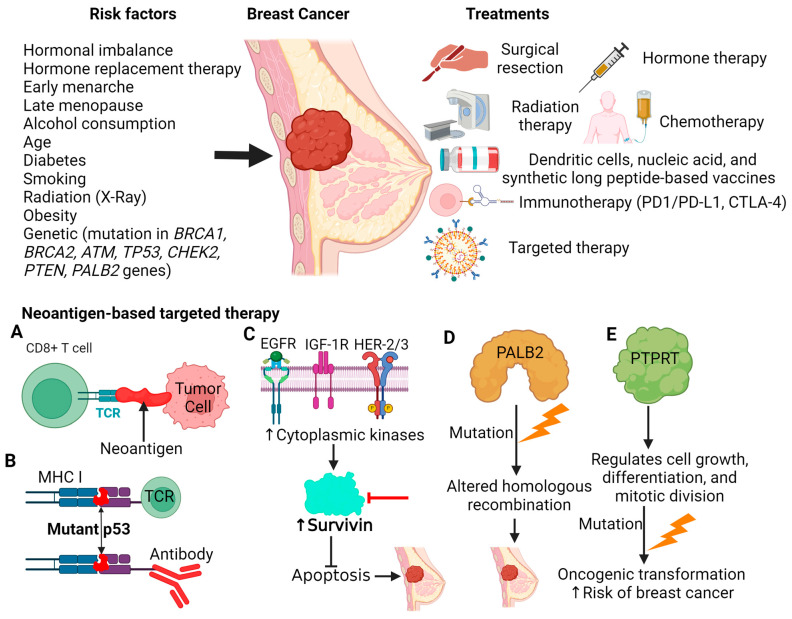
Risk factors, available treatment, and neoantigen-based targeted therapy. Risk factors including age, obesity, diabetes, exposure to radiation, hormone replacement therapy, early menarche, late menopause, alcohol consumption, smoking, and genetic predisposition due to mutation in various genes increases the risk of breast cancer. Surgery, radiation, hormone therapy, chemotherapy, and targeted therapies are currently available therapies for breast cancer depending on whether the cancer is localized or advanced. Immunotherapy involves targeting a specific target to treat the cancer and the altered proteins formed due to mutations are targeted. Neoantigen-based therapy involves targeting a tumor-associated specific neoantigen to decrease tumor size by either attenuating the tumor cell proliferation or increasing tumor cell death. (**A**) MHC-bound neoantigen presented to the T cell surface interacts with and stimulates a CD8+ T cell to attenuate tumor growth. Mutation in TP53 (**B**), survivin (**C**), PALB2 (**D**), and PTPRT (**E**) results in an increased risk of breast cancer. Epidermal growth factor receptor (EGFR), insulin-like growth factor 1 receptor (ICF-1R), human epidermal growth factor receptor 2/3 (HER2/3).

**Table 1 antibodies-13-00046-t001:** Clinical trials of neoantigen-based therapies for breast cancer. Trial information was collected from Clinicaltrials.gov by accessed on 5 October 2023; Conditions/disease: “Breast Cancer” and other terms: “Neoantigen”. All studies were included (*n* = 13).

Trials ID/Phae	Intervention/Treatment	Breast Cancer Subtype	Enrollment	Status/Results	Sponsor
NCT03199040 Phase 1	Drug: Durvalumab Biological: Neoantigen DNA vaccine Device: TDS-IM system (Inchor Medical Systems) Procedure: Peripheral blood draw	Triple Negative Breast Cancer	18 (Actual)	Terminated—Drugs/equipment unavailable and insufficient funding	Washington University School of Medicine
NCT04105582 Phase 1	Biological: Neo-antigen pulsed Dendritic cell	Triple Negative Breast Cancer	5 (Actual)	Completed—No results posted	Universidad Nacional de Colombia
NCT03606967 Phase 2	Procedure: Biopsy Procedure: Biospecimen Collection Drug: Carboplatin	Triple Negative Breast Cancer	70 (Estimated)	Recruiting	National Cancer Institute (NCI)
NCT05576077 Phase 1	Biological: TBio-4101 Drug: Pembrolizumab	Triple Negative Breast Cancer	60 (Estimated)	Recruiting	Turnstone Biologics, Corp.
NCT03970382 Phase 1	Biological: NeoTCR-P1 adoptive cell therapy Biological: Nivolumab Biological: IL-2	Solid Tumor	21 (Actual)	Suspended—Business decision	PACT Pharma, Inc.
NCT05269381 Phase 1	Drug: Cyclophosphamide Biological: Neoantigen Peptide Vaccine Biological: Pembrolizumab Biological: Sargramostim	Triple Negative Breast Cancer	36 (Estimated)	Recruiting	Mayo Clinic
NCT05098210 Phase 1	Biological: Neoantigen Peptide Vaccine Biological: Nivolumab Drug: Poly ICLC	Hormone Receptor Positive Her2 Negative Metastatic Refractory Breast Cancer	20 (Estimated)	Recruiting	Fred Hutchinson Cancer Center
NCT03289962 Phase 1	Drug: Autogene cevumeran Drug: Atezolizumab	Triple Negative Breast Cancer	272 (Actual)	Active, not recruiting	Genentech, Inc.
NCT03552718 Phase 1	Biological: YE-NEO-001	Triple Negative Breast Cancer	16 (Estimated)	Active, not recruiting	NantBioScience, Inc.
NCT03361800 Early Phase 1	Drug: Entinostat	Stage I-IIIC, Hormone Receptor—Positive (HR+) or Triple Negative Breast Cancer (TNBC)	5 (Actual)	Terminated—Funding withdrawn.	UNC Lineberger Comprehensive Cancer Center
NCT02883062 Phase 2	Drug: Atezolizumab Procedure: Biospecimen Collection Drug: Carboplatin Procedure: Lumpectomy Procedure: Mastectomy Drug: Paclitaxel	Stage II-III Triple-Negative Breast Cancer	67 (Actual)	Active, not recruiting	National Cancer Institute (NCI)
NCT03409198 Phase 2	Drug: Ipilimumab Drug: Nivolumab Drug: Pegylated liposomal doxorubicin Drug: Cyclophosphamide	Metastatic Hormone Reseptor Positive Breast Cancer	82 (Actual)	Completed—No results posted	Oslo University Hospital
NCT06281860 Phase 1	Drug: Cisplatine Teva^®^	Carcinoma Breast Stage IV	39 (Estimated)	Recruiting	Dr Jean Yannis PERENTES

## Data Availability

This is a review article and all data collected from the research articles has been included in the manuscript.

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
