# Peer review of "Targeting Neoantigens in Cancer: Possibilities and Opportunities in Breast Cancer"

_2073-4468, 2024, doi:10.3390/antib13020046_

Round 1

Reviewer 1 Report

Comments and Suggestions for Authors

Chaudhry et al has provided a comprehensive review about breast cancer treatment with a focus on targeting neoantigens. The manuscript includes basic introduction, highlights some clinical trial examples, and pinpoint the limitations and future directions of neoantigens targeted therapy. It is well written, and could be a good resource for the field. However, some minor points should be addressed before publication.

1.     Line 75. The major issue of surgical intervention is the advanced stage of the disease, which makes it incurable by simple surgical removal, rather than physiological and psychosocial risks.

2.     In section 2, starting from line 85, the authors are recommended to explicitly point out that the efficacy of ICB is very limited in breast cancer, unlike lung cancer and melanoma. This is not only an important fact of the current situation, but also makes a smooth transition for the introduction of neoantigen therapy.

3.     In the future direction section, the authors should discuss more about the combination of neoantigen targeting with either ICB or CART. The current strategy to boost ICB efficiency in breast cancer is to combine it with therapy tarting DNA damage repair pathway to elevate immune response, however, this has suffered severe toxicity. Similarly, CAR T cells targeting common markers like CD20 also facing strong side effects like cytokine storm. Whether neoantigen has the potential to alleviate this dilemma by providing a more unique target is worth mentioning.

Author Response

Comment: Chaudhry et al has provided a comprehensive review about breast cancer treatment with a focus on targeting neoantigens. The manuscript includes basic introduction, highlights some clinical trial examples, and pinpoint the limitations and future directions of neoantigens targeted therapy. It is well written, and could be a good resource for the field. However, some minor points should be addressed before publication.

Response: Thank you for your comments and suggestions.

Concern 1: Line 75. The major issue of surgical intervention is the advanced stage of the disease, which makes it incurable by simple surgical removal, rather than physiological and psychosocial risks.

Response: We thank the reviewer for the recommendation. We have modified the text in the revised manuscript.

Concern 2: In section 2, starting from line 85, the authors are recommended to explicitly point out that the efficacy of ICB is very limited in breast cancer, unlike lung cancer and melanoma. This is not only an important fact of the current situation, but also makes a smooth transition for the introduction of neoantigen therapy.

Response: We thank the reviewer for the suggestion. We have modified the text in the revised manuscript.

Concern 3: In the future direction section, the authors should discuss more about the combination of neoantigen targeting with either ICB or CART. The current strategy to boost ICB efficiency in breast cancer is to combine it with therapy tarting DNA damage repair pathway to elevate immune response, however, this has suffered severe toxicity. Similarly, CAR T cells targeting common markers like CD20 also facing strong side effects like cytokine storm. Whether neoantigen has the potential to alleviate this dilemma by providing a more unique target is worth mentioning.

Response: We thank the reviewer for this suggestion and have now dedicated a portion of the Future Direction section towards extrapolating on the potential that CAR T cells harness in treating solid tumors associated with breast cancer. We briefly explain the mechanism of CAR T cells, their limitations, and weave in the possible benefit of co administering neoantigen based therapies.

Reviewer 2 Report

Comments and Suggestions for Authors

To the authors:

Chaudhry et al. provided a comprehensive summary of current research on identifying neoantigens and utilizing such neoantigens as direct targets for therapy, known as neoantigen-based immunotherapy. The authors discussed the limitations of standard treatments due to therapeutic resistance, emphasizing the need to explore new treatment options for resistant metastatic breast cancer. Neoantigen- or tumor-associated antigen-based immunotherapy is a promising strategy for the treatment of cancer and has been discussed in the following review. The manuscript is well-written, with an introductory section presenting general findings followed by an extensive review of neoantigen-based immunotherapies.

Minor Concern:

Line 62: “Chemotherapy, surgery, and radiation therapy”, shouldn’t it be the order  “Surgery, radiation -, and chemotherapy”?

Line 92: “presented in the MHC of APCs”, needs to be more precise, such as this “presented on MHC of APCs” or “neo-antigenic peptides bond in the binding groove of MHC of APCs are presented to T cells”

Line 117: It is not clear how T cell immunity is responsible for “resistance”, “while neoantigens play a critical role in the efficacy of PD-1/PD-1L checkpoint inhibitors, resistance can also occur due to neoantigens loss resulting from selective clearance of neoantigen-presenting cells by T-cell immunity”. Are CD8+ T cells responsible for “killing” neoantigen presenting APCs?

Line 143: “Neoantigen refers to antigens that result from somatic mutations of major histocompatibility (MHC) molecules of tumor cells distinguishing them from self-antigens.” The MHC molecules are not “mutated”; instead, the neoantigens. This needs to be corrected. In addition, it should be stated that antigens/neoantigens are processed intracellularly (antigen processing), and antigenic peptides are bound to MHC molecules. Because in line 148, it is referred to as “mutant peptides”.

Line 190: “…the overall reduced expression of genes or loss of the mutant allele in melanoma…”, this should be more precise.

Line 193: “…a few immune cells…”, should be more precise, and the expression “few” is not appropriate.

Line  203: The “…loss of diversity…”. The paragraph needs to be rewritten and clearer.

Line 210/215: Cancer, lower-case letter “c” or upper-case letter? Need to be consistent.

Line 228: The sentence does not represent the statement from the paper from Umano et al.; please revise

Line 339: Dendritic cells, does not fit into the row.

Line 374: “haplotype” and in the next sentence it was referred to “ MHC alleles”?

Line 405: Keep consistent, upper or lower case letter, “(iii) Analysis” compared to “(v) variant”

Line 422: Should be MHC class I

Line 432: Being consistent, MHC class I, lower case “c in class”

Additional comments:

1. How (or is it possible) can a general vaccine be generated considering the individual MHC alleles, peptide binding to MHC, antigen processing, or different antigen-presenting cells?

2. Comment: 5. Neoantigens and 6. TP53 tumor-associated antigen etc., it would be clearer to categorize 5. tumor-specific antigens (neoantigens) and 6. tumor-associated antigens, followed by subheadings/examples , such as TP53, etc. 

3. Table 1: Figure legend refers to “Clinical Trials of Neoantigen-based therapies for Breast Cancer”. The table 1, should include which kind of neoantigens were used. TB53 or PALB2?

Comments on the Quality of English Language

Minor corrections.

Author Response

Comments: Chaudhry et al. provided a comprehensive summary of current research on identifying neoantigens and utilizing such neoantigens as direct targets for therapy, known as neoantigen-based immunotherapy. The authors discussed the limitations of standard treatments due to therapeutic resistance, emphasizing the need to explore new treatment options for resistant metastatic breast cancer. Neoantigen- or tumor-associated antigen-based immunotherapy is a promising strategy for the treatment of cancer and has been discussed in the following review. The manuscript is well-written, with an introductory section presenting general findings followed by an extensive review of neoantigen-based immunotherapies.

Response: Thank you for your comments.

Concern 1: Line 62: “Chemotherapy, surgery, and radiation therapy”, shouldn’t it be the order  “Surgery, radiation -, and chemotherapy”?

Response: Thank you for your comment. We have modified the text in the revised manuscript.

Concern 2: Line 92: “presented in the MHC of APCs”, needs to be more precise, such as this “presented on MHC of APCs” or “neo-antigenic peptides bond in the binding groove of MHC of APCs are presented to T cells”

Response: Thank you for your comment. We have modified the text in the revised manuscript.

Concern 3: Line 117: It is not clear how T cell immunity is responsible for “resistance”, “while neoantigens play a critical role in the efficacy of PD-1/PD-1L checkpoint inhibitors, resistance can also occur due to neoantigens loss resulting from selective clearance of neoantigen-presenting cells by T-cell immunity”. Are CD8+ T cells responsible for “killing” neoantigen presenting APCs?

Response: Thank you for your question. We corrected this statement as “resistance can occur due to an active antitumor response resulting in immunoselection and outgrowth of neoantigen-loss variants and tumor immune escape” as the literature source does not list T cells as responsible for “killing: neoantigen presenting APCs.

Concern 4: Line 143: “Neoantigen refers to antigens that result from somatic mutations of major histocompatibility (MHC) molecules of tumor cells distinguishing them from self-antigens.” The MHC molecules are not “mutated”; instead, the neoantigens. This needs to be corrected. In addition, it should be stated that antigens/neoantigens are processed intracellularly (antigen processing), and antigenic peptides are bound to MHC molecules. Because in line 148, it is referred to as “mutant peptides”.

Response: Thank you for your suggestions. We have modified the text accordingly in the revised manuscript.

Concern 5: Line 190: “…the overall reduced expression of genes or loss of the mutant allele in melanoma…”, this should be more precise.

Response: Thank you for your suggestions. We have modified the text accordingly in the revised manuscript.

Concern 6: Line 193: “…a few immune cells…”, should be more precise, and the expression “few” is not appropriate.

Response: Thank you for your suggestions. We have modified the text accordingly in the revised manuscript.

Concern 7: Line  203: The “…loss of diversity…”. The paragraph needs to be rewritten and clearer.

Response: Thank you for your suggestions. We have modified the text accordingly in the revised manuscript.

Concern 8: Line 210/215: Cancer, lower-case letter “c” or upper-case letter? Need to be consistent.

Response: Thank you for your suggestions. We have modified the text accordingly in the revised manuscript.

Concern 9: Line 228: The sentence does not represent the statement from the paper from Umano et al.; please revise

Response: Thank you for your suggestions. We have modified the text accordingly in the revised manuscript.

Concern 10: Line 339: Dendritic cells, does not fit into the row.

Response: Thank you for your suggestions. We have modified the text accordingly in the revised manuscript.

Concern 11:Line 374: “haplotype” and in the next sentence it was referred to “ MHC alleles”?

Response: Thank you for your suggestions. We have modified the text accordingly in the revised manuscript.

Concern 12: Line 405: Keep consistent, upper or lower case letter, “(iii) Analysis” compared to “(v) variant”

Response: Thank you for your suggestions. We have modified the text accordingly in the revised manuscript.

Concern 13: Line 422: Should be MHC class I

Response: Thank you for your suggestions. We have modified the text accordingly in the revised manuscript.

Concern 14: Line 432: Being consistent, MHC class I, lower case “c in class”

Response: Thank you for your suggestions. We have modified the text accordingly in the revised manuscript.

Concern 15: How (or is it possible) can a general vaccine be generated considering the individual MHC alleles, peptide binding to MHC, antigen processing, or different antigen-presenting cells?

Response: We thank you for this question as it possesses a serious concern with regards to the mass production of neoantigen based vaccines for the treatment of different forms of breast cancer. Due to the individual differences that exist in MHC alleles person-to-person the current process of creating vaccines is prolonged and dependent upon bioinformatic algorithms to correctly identify immunogenic neoantigens. As a result, this process generates profound costs that make vaccines unaffordable for the vast majority of patients as alluded to within the Future Direction section. Throughout our literature searches we have solely encountered data based on studies of personalized vaccines. While the advent of general vaccines will likely minimize the cost and even the time required to begin treatment, current studies instead focus on streamlining existing processes for highly specific vaccines. Some additional means of enhancing the production of personalized vaccines include enhancement of current NGS technology and mass spectrometry, as improving the two methods will enable researchers to find new neoantigens; we have since included these concepts within our revision.

Comment: 5. Neoantigens and 6. TP53 tumor-associated antigen etc., it would be clearer to categorize 5. tumor-specific antigens (neoantigens) and 6. tumor-associated antigens, followed by subheadings/examples , such as TP53, etc. 

Response: Thank you for your comment and suggestions. The neoantigens discussed in these sections are all under the category of tumor specific antigens. We have edited the title of this section to be more specific.

Table 1: Figure legend refers to “Clinical Trials of Neoantigen-based therapies for Breast Cancer”. The table 1, should include which kind of neoantigens were used. TB53 or PALB2?

Response: Thank you for your comment. We checked again at Clinicaltrials.gov for information regarding the specific neoantigen that was used, however, the information only on the device/treatment that was used has been provided.

Reviewer 3 Report

Comments and Suggestions for Authors

In the manuscript “Targeting Neoantigens in Cancer: Possibilities and Opportunities in Breast Cancer,” the authors reviewed neoantigens and their potential use in breast cancer treatment. This is an interesting topic that contributes to knowledge in the area, but certain issues must be reviewed.

Major revisions

1.    Authors must mention in the abstract and introduction the gap that the manuscript will fill in the current knowledge, the objective, and the possible conclusion of the manuscript.

2.    The authors must specify what type of cancer they are talking about because it is common for them to talk about cancer, but they do not specify if it is breast or some other type of cancer. Even when talking about breast cancer, they must specify what type of breast cancer it is.

3.    Each section must be improved by adding an explanation at the beginning of their description.

4.     Each section must have concluded. The author must resume and conclude each section: for instance, the data above suggest… or the information concluded that.

5.    Define all the abbreviations presented in the text and do so in their first appearance, either in the abstract or the body of the manuscript. For example, AKT, PI3K, etc.

6.    The authors should add figures on the use of neoantigens in breast cancer.

7.    In many sections, cancer is discussed in a general way; however, no emphasis is placed on which type of cancer. Cancer itself is heterogeneous, and breast cancer is even more heterogeneous, so making these generalities does not meet the objective of the manuscript, which deals with breast cancer, not cancer in general.

8.    Authors must specify what type of study model was used in the reviewed works in the manuscript.

9.    The authors must talk about breast cancer and its subtypes. Taking it as a cancer without subtypes is not a good approach because the processes can change from subtype to subtype.

10. The authors must develop a conclusion from their manuscript associating neoantigens with breast cancer heterogeneity.

11. The authors should point out the limitations of the use of neantigens in the possible treatment of breast cancer.

Comments on the Quality of English Language

No comments

Author Response

Comment: In the manuscript “Targeting Neoantigens in Cancer: Possibilities and Opportunities in Breast Cancer,” the authors reviewed neoantigens and their potential use in breast cancer treatment. This is an interesting topic that contributes to knowledge in the area, but certain issues must be reviewed.

Response: Thank you for your comments.

Concern 1.    Authors must mention in the abstract and introduction the gap that the manuscript will fill in the current knowledge, the objective, and the possible conclusion of the manuscript.

Response: We thank the reviewer for their suggestion and have since added a concluding sentence within the abstract detailing the gap in logic that this paper seeks to address.

Concern 2: The authors must specify what type of cancer they are talking about because it is common for them to talk about cancer, but they do not specify if it is breast or some other type of cancer. Even when talking about breast cancer, they must specify what type of breast cancer it is.

Response: We thank the reviewer for the recommendation. We have included the type of breast cancer in table 1, as suggested, in the revised manuscript.

Concern 3: Each section must be improved by adding an explanation at the beginning of their description.

Response: Thank you for your suggestion. We have revised the manuscript as per your suggestion.

Concern 4. Each section must have concluded. The author must resume and conclude each section: for instance, the data above suggest… or the information concluded that.

Response: Thank you for your suggestion. We have revised the manuscript as per your suggestion.

Concern 5.    Define all the abbreviations presented in the text and do so in their first appearance, either in the abstract or the body of the manuscript. For example, AKT, PI3K, etc.

Response: Thank you for your suggestion. We have revised the manuscript as per your suggestion.

Concern 6.    The authors should add figures on the use of neoantigens in breast cancer.

Response: Thank you for your suggestion. We have included Figure 1 in the revised manuscript.

Concern 7.    In many sections, cancer is discussed in a general way; however, no emphasis is placed on which type of cancer. Cancer itself is heterogeneous, and breast cancer is even more heterogeneous, so making these generalities does not meet the objective of the manuscript, which deals with breast cancer, not cancer in general.

Response: We thank the reviewer for the recommendation. We have revised the manuscript as per your suggestion.

Concern 8.    Authors must specify what type of study model was used in the reviewed works in the manuscript.

Response: Thank you for your comment. We included original and review articles in this study and we have mentioned this in the revised manuscript.

Concern 9.    The authors must talk about breast cancer and its subtypes. Taking it as a cancer without subtypes is not a good approach because the processes can change from subtype to subtype.

Response: We thank the reviewer for the recommendation. We have revised the manuscript as per your suggestion.

Concern 10. The authors must develop a conclusion from their manuscript associating neoantigens with breast cancer heterogeneity.

Response: Thank you for your suggestions. We have revised the conclusion.

Concern 11. The authors should point out the limitations of the use of neoantigens in the possible treatment of breast cancer.

Response: Many of the limitations seen in neoantigens as a means of treating cancer are mirrored within breast cancer. However, we have since included a specific study exploring the limitations arising from low sensitivity and specificity in selecting for neoantigens.

Round 2

Reviewer 3 Report

Comments and Suggestions for Authors

Authors must describe Figure 1 in both the text and the title of Figure 1. For example, the BRCA1 and BRCA2 genes are mentioned where Figure 1 is first referenced, but these genes do not appear in Figure 1. The treatments where Table 1 is referenced again are not mentioned either. The authors must make an effort to make a figure corresponding to the text and not limit themselves to adding random drawings.

Although the authors highlight the different types of breast cancer in Table 1, they do not mention them in the text, and this remains a major flaw of the manuscript. It is clear that the proposed treatment will work for some subtypes, and for others, it will not since the subtypes of breast cancer are diametrically different. Authors have to make an effort to clarify this so that their manuscript is more interesting to the reader.

Comments on the Quality of English Language

no comments

Author Response

Comment 1: Authors must describe Figure 1 in both the text and the title of Figure 1. For example, the BRCA1 and BRCA2 genes are mentioned where Figure 1 is first referenced, but these genes do not appear in Figure 1.

Response: Thank you for your comments and suggestions. We have modified Figure 1, discussed the figure in the figure legend, and included the relevant text in the figure and vice-versa.

Comment 2 The treatments where Table 1 is referenced again are not mentioned either. The authors must make an effort to make a figure corresponding to the text and not limit themselves to adding random drawings.

Response: Thank you for your comment and suggestion. We have modified the text in the revised manuscript and changed the table's location, placing it in a suitable relevant position.

Comment 3: Although the authors highlight the different types of breast cancer in Table 1, they do not mention them in the text, and this remains a major flaw of the manuscript. It is clear that the proposed treatment will work for some subtypes, and for others, it will not since the subtypes of breast cancer are diametrically different. Authors have to make an effort to clarify this so that their manuscript is more interesting to the reader.

Response: Thank you for your comments and suggestions. We have revised the text accordingly and have included the text emphasizing the role of tumor resistance, and the role of cancer stem cells. Tumor heterogeneity in treatment ineffectiveness and the need for personalized therapy by targeting neoantigens.